# The Relationship between Diffusion-Weighted Magnetic Resonance Imaging Lesions and 24-Hour Rhythm Holter Findings in Patients with Cryptogenic Stroke

**DOI:** 10.3390/medicina55020038

**Published:** 2019-02-04

**Authors:** Muhammet Gürdoğan, Sezgin Kehaya, Selçuk Korkmaz, Servet Altay, Uğur Özkan, Çağlar Kaya

**Affiliations:** 1Department of Cardiology, School of Medicine, Trakya University, 22030 Edirne, Turkey; svtaltay@gmail.com (S.A.); drugurozkan@hotmail.com (U.Ö.); caglarkaya2626@gmail.com (Ç.K.); 2Department of Neurology, School of Medicine, Trakya University, 22030 Edirne, Turkey; sezginkehaya@yahoo.com; 3Department of Biostatistics and Medical Informatics, School of Medicine, Trakya University, 22030 Edirne, Turkey; selcukorkmaz@gmail.com

**Keywords:** cryptogenic stroke, cardioembolism, premature atrial contraction, atrial run

## Abstract

*Background and objectives:* Cranial magnetic resonance imaging findings of patients considered to be cryptogenic stroke may be useful in determining the clinical and prognostic significance of arrhythmias, such as atrial premature beats and atrial run attacks, that are frequently encountered in rhythm Holter analysis. This study was conducted to investigate the relationship between short atrial runs and frequent premature atrial contractions detected in Holter monitors and infarct distributions in cranial magnetic resonance imaging of patients diagnosed with cryptogenic stroke. *Materials and Methods:* We enrolled the patients with acute ischemic stroke whose etiology were undetermined. We divided the patients in two groups according to diffusion-weighted magnetic resonance imaging as single or multiple vascular territory acute infarcts. The demographic, clinical, laboratory, echocardiographic, and rhythm Holter analyses were compared. *Results:* The study investigated 106 patients diagnosed with cryptogenic stroke. Acute cerebral infarctions were detected in 31% of the investigated patients in multiple territories and in 69% in a single territory. In multivariate logistic regression analysis, the total premature atrial contraction count (OR = 1.002, 95% CI: 1.001–1.004, *p* = 0.001) and short atrial run count (OR = 1.086, 95% CI: 1.021–1.155, *p* = 0.008) were found as independent variables that could distinguish between infarctions in a single or in multiple vascular territories. *Conclusions:* Rhythm Holter monitoring of patients with infarcts detected in multiple vascular territories showed significantly higher premature atrial contractions and short atrial run attacks. More effort should be devoted to the identification of cardioembolic etiology in cryptogenic stroke patients with concurrent acute infarcts in the multiple vascular territories of the brain.

## 1. Introduction

Stroke, one of the leading causes of mortality and morbidity worldwide, is the third most common cause of mortality following ischemic heart disease and cancer and is the most common cause of long-term disability, especially in people over 60 years of age [1,2]. In the literature, 81.7% of stroke cases are reported to be ischemic, 12.4% of them are due to intracerebral hemorrhage, and 2.9% of them are due to subarachnoid hemorrhage [3]. The most common etiologic causes of ischemic stroke are classified as atherosclerosis of major arteries, cardioembolism, small vessel occlusions, other determined causes, and undetermined causes [4]. Cardioembolic stroke accounts for 20–30% of the cases, whereas no etiologic cause is found in 30–40% of the patients despite having performed all the necessary diagnostic tests [5,6]. The etiology of cryptogenic strokes is frequently reported as cardiac thromboembolism. Atrial fibrillation is a well-known etiologic cause of cardioembolic stroke. In half of cardioembolic strokes, it was determined that multiple vascular territories were involved in magnetic resonance examinations [6,7]. The role of premature atrial contractions and short atrial runs, which is known to trigger atrial fibrillation, is not clear in stroke etiology.

This study was carried out to determine whether there is a relationship between the distribution of acute cerebral infarcts in the cranial diffusion-weighted magnetic resonance image of the patients diagnosed with stroke of undetermined etiology and the demographic, clinical, laboratory, echocardiographic, and rhythm Holter monitoring data of the patients. 

## 2. Materials and Methods

### 2.1. Subject Selection

The study retrospectively evaluated the data of 244 patients for whom 24-hour rhythm Holter monitoring was requested in order to investigate the cardioembolic etiology in cryptogenic stroke between April 2015 and April 2018. Patients with a low left ventricular ejection fraction (LVEF ≤ 30%), atrial fibrillation, rheumatic mitral stenosis and mechanical valve, spontaneous echo contrast or thrombus in the left atrium or the left atrial appendage, aortic complicated atherosclerotic plaques, diagnosed patent foramen ovale, diagnosed secondary stroke of atherosclerotic etiology, or hemorrhagic stroke, without cranial diffusion-weighted magnetic resonance imaging (DWI-MRI) and echocardiographic examination data, who were also not assessed as healthy or had under 12 hours of rhythm monitoring and were 18 years of age were excluded. (Figure 1) The approval of the Scientific Research Ethics Committee of the Faculty of Medicine of Trakya University (TUTF-BAEK 2018/207) was obtained for the research.

### 2.2. Collection of Data

Sociodemographic, clinical, laboratory, echocardiographic, and rhythm Holter monitorization data and cranial DWI-MR imaging data were recorded for the patients included in the study. Rhythm analysis of all patients was performed with a 12-lead, 24-hour, ambulatory ECG (Cardioscan 11 Premier Holter System, DMS, Stateline, NV, USA). The records were transferred to the computer environment and evaluated initially with the Holter program (Cardioscan-11 Holter ECG Systems, DMSoftware), and then they were visually examined, and the areas with noise were excluded from the evaluation. Accelerated atrial beats of 3 or more, or less than 30 seconds, were defined as a short atrial run [7]. (Figure 2A,B)

Echocardiography (Vivid 7® GE Medical System, Horten, Norway) evaluations were done for all of the patients, LVEF values were measured using the Simpson method from the apical four-chamber view, and the left atrial diameters were measured by M-mode echocardiography at a distance from the anterior margin to posterior margin in the parasternal long axis window in accordance with the guidelines of the American Echocardiography Society [8].

### 2.3. MR Protocol and Evaluation

The 1.5 Tesla DWI-MR images of patients who were hospitalized with acute stroke and considered as stroke with undetermined origin based on the results of the examinations were retrospectively analyzed, and patients with an acute infarction in single and multiple vascular territories were divided into two groups (Figure 3A–D). Multiple vascular territory acute infarcts are known to be associated with cardioembolism [9]. Single vascular territory infarcts are classified as unilateral anterior or posterior circulation infarcts. Multiple vascular territory infarcts were defined as bilateral anterior or anterior and posterior circulation concurrent infarction. The cranial time-of-flight MR angiography (TOF-MRA) of patients with acute multiple vascular territory infarcts in DWI-MRI were reviewed for arterial variations. Patients with bilateral hemispheric circulation with a single carotid system that could lead to a bilateral anterior circulation infarct or patients with a carotid-origin posterior cerebral artery (fetal-type posterior cerebral artery) that could lead to an anterior and posterior circulation infarct were excluded from the study because they could not be classified.

### 2.4. Statistical Examinations

A power analysis was performed with the effect size 0.5, and based on the study findings conducted by [9] and [10], a 20% type II error rate and 5% significance level were also applied. As a result, 102 subjects were sufficient to investigate the relationship between diffusion-weighted magnetic resonance imaging lesions and 24-hour rhythm Holter findings in patients with cryptogenic stroke. SPSS 20.0 (IBM SPSS Statistics for Windows, version 20.0; Armonk, NY: IBM Corp. USA) was used for the analysis of the data. The normal distribution assumption was checked using the Shapiro–Wilk test before comparing the quantitative variants. For the variables with normal distribution assumption, group comparisons were performed by independent two-sample *t*-tests and by the Mann–Whitney *U* test for variables without a normal distribution assumption. The qualitative variables were compared using the Chi-square test. Both univariate and multivariate logistic regression analyses were performed to determine the factors that could distinguish between single and multiple vascular territories. The mean and standard deviation were given for quantitative variables with normal distribution, while the median and quartiles (25th–75th percentile) were given for quantitative variables without a normal distribution. The frequency and percentage were given for qualitative variables. A value of *p* < 0.05 was accepted as the statistical significance limit.

## 3. Results 

The flow chart of the study sample recruitment is shown in Figure 1. Finally, cranial MR imaging of 106 patients (68% male, 32% female) with a diagnosis of cryptogenic stroke was examined in the study. In 31% of patients, infarcts were detected in multiple vascular territories, whereas in 69% of patients, infarcts were found in a single vascular territory. The differences in demographic, clinical, laboratory, echocardiographic, and 24-hour rhythm Holter data were compared between the two groups. The characteristics of the patients and all the comparison results are summarized in Table 1.

According to the results in Table 1, the age (65.12 ± 12.79) of patients with infarcts in multiple vascular territories was significantly higher than in patients with infarcts in a single vascular territory (59.07 ± 14.21) (*p* = 0.039). The frequency of hypertension (*p* = 0.041), C-reactive protein level (*p* = 0.037), total premature atrial contractions (PAC) (*p* < 0.001), atrial run count (*p* < 0.001), and PACs/total heart rate (*p* < 0.001) in patients with infarcts in multiple vascular territories were found to be significantly higher than in patients with infarcts in the single vascular territory (*p* < 0.001). The hemoglobin level of patients with infarcts in multiple vascular territories was found to be significantly lower than in patients with infarcts in a single vascular territory ( *p*< 0.001). (Table 2)

To determine the factors that can discriminate between single and multiple vascular territories, univariate logistic regression analyses were performed for all variables with significant differences between the two groups. In univariate logistic regression analysis, variables with *p* < 0.20 were included in the multivariate logistic regression model. The variables involved in multivariate logistic regression analysis were age (odds ratio (OR) = 1.034, *p* = 0.042), mean heart rate (OR = 1.033, *p* = 0.117), C-reactive protein (CRP) (OR = 1.175, *p* = 0.032), hemoglobin (OR = 0.689, *p* = 0.003), total PAC count (OR = 1.003, *p* < 0.001), short atrial run count (OR = 1.126, *p* < 0.001), CHA_2_DS_2_-VASc score (OR = 1.246, *p* = 0.058), hypertension (OR = 2.424, *p* = 0.043), diabetes mellitus (OR = 2.094, *p* = 0.094), and hyperlipidemia (OR = 2.046, *p* = 0.096) (Table 3).

As a result of multivariate logistic regression analysis, total PAC count (OR = 1.002, *p* = 0.001) and short atrial run count (OR = 1.086, *p* = 0.008) were found as independent variables that could discriminate between infarctions in single and multiple vascular territories (Table 4).

## 4. Discussion

The main findings of this study can be summarized as follows: i) rhythm Holter monitoring of patients with infarcts detected in multiple vascular territories showed significantly higher premature atrial contractions and short atrial run attacks compared to patients with infarcts in a single vascular territory, ii) the mean age of patients with infarcts in multiple vascular territories and the frequency of hypertension were higher in these patients, iii) CRP levels were higher and hemoglobin levels were lower in patients with infarcts in multiple vascular territories during hospitalization.

Rhythm Holter monitoring data of patients who were considered as cryptogenic stroke cases and thus treated according to current guidelines revealed frequent premature atrial contractions and short atrial run attacks in a remarkable portion of patients [11]. In the cranial MR scans of the patients included in the study, there was a statistically significant difference in the frequency of premature atrial contraction and short atrial run in rhythm Holter monitorization between the two groups that were separated by detection of infarctions in single or multiple cerebral vascular territories (*p* < 0001).

Despite extensive etiologic investigations, no specific cause has been found in 30–40% of ischemic stroke patients [5]. In the literature, it has been reported that the etiology of these patients with strokes of undetermined origin, or otherwise referred to as "cryptogenic strokes," is likely to be thromboembolic, and the emboli are mostly of cardiac origin [12,13]. The etiologic cause is reported to be cardioembolic in half of the cases who had stroke and underwent cranial MR examinations and had infarcts in multiple vascular territories [14,15]. In multiple vascular territories observed to form infarcts at similar times in cases with cardioembolic etiology, acute infarcts on both the right and left hemispheres or both the anterior and posterior circulation provide clues [16]. In our study, 31% of our patients had infarcts in multiple vascular territories.

Cardioembolic stroke has a worse prognosis than other etiological factors in terms of mortality, disability, and recurrence [17]. Although many cardioembolic risk factors that result in ischemic stroke are known, the most common cardioembolic stroke cause in daily clinical practice is atrial fibrillation [5,17]. Paroxysmal or chronic atrial fibrillation is known as a well-defined etiologic cause of ischemic stroke, and the risk of stroke can be reduced by appropriate anticoagulant therapy with both primary and secondary prevention [18]. However, it is not clear whether premature atrial contractions and short atrial runs, which are known to trigger atrial fibrillation and are frequently encountered in 24-hour rhythm Holter monitoring of stroke patients, are associated with the pathophysiology of stroke and therefore whether treatment is required [18,19,20,21,22,23,24,25]. It is not known whether these arrhythmias pose a risk for stroke alone before affecting the atrial tissue and entering the process of turning into atrial fibrillation over time. However, in our study findings, in the rhythm monitorization analysis of patients with infarcts in multiple vascular territories, it was observed that premature atrial contractions and short atrial runs were observed at a significant level compared to the patients with infarcts in single vascular territory. This is consistent with literature knowledge that suggests that possible paroxysmal atrial fibrillation attacks may not be detectable with the rhythm Holter monitoring, and that emphasizes the need for long-term rhythm monitoring [21,22,23].

In our study, the analysis of patients who had infarcts in multiple or single vascular territories had similar rhythm Holter monitoring durations of over 23 hours. In updated guidelines, ECG rhythm Holter monitoring is recommended for 24 hours or longer in investigating cardioembolic etiology, although the most effective monitoring duration is not exactly determined [25,26]. Gladstone et al. reported that a significant number of more atrial fibrillation episodes could be detected with noninvasive, thirty-day monitoring in patients with cryptogenic stroke or transient ischemic attacks, and oral anticoagulant therapy was thus started twice. [27]. Although our current knowledge does not support the need for the administration of prophylactic anticoagulant therapy because of the uncertainty of its role in cryptogenic stroke etiology, patients with frequent premature atrial contractions and short atrial runs need to be closely monitored in terms of risk of conversion to paroxysmal or chronic atrial fibrillation because it is the most important cardioembolic cause of stroke [28].

There were no significant differences between patients with single and multiple infarcts in terms of gender, LVEF, left atrial dimension, CHA2DS2-VASc score, diabetes mellitus, hyperlipidemia, and coronary artery disease presence in cranial MR examinations. However, it was found that the mean age of the patients who had multiple infarcts was higher than the other group. This finding is consistent with previous studies showing that the severity and extent of stroke was increased with age [29,30]. The most common risk factor encountered for stroke is hypertension [31,32]. The presence of hypertension is reported to be one of the factors increasing stroke severity [33]. It was observed in our study that a history of hypertension was significantly reported more in patients with infarcts in multiple vascular territories.

Increased blood CRP levels have been shown to be associated with an increased risk of stroke in the general population [34]. Moreover, elevated blood CRP levels have been reported to correlate with the severity of neurological deficits to form and the mortality after stroke [35,36]. In light of this literature, it is not surprising that the CRP level is higher in patients with infarcts in multiple vascular territories. 

Anemia (defined as hemoglobin levels <12.0 g/dL in women and <13.0 g/dL in men according to World Health Organization criteria) is a common clinical condition in patients with stroke and has been reported in the literature in up to 30% of stroke patients [37,38]. In a large-scale analysis by Barlas et al., the severity of stroke and the mortality rate in one year were higher in the presence of anemia [39]. Our study also found that the frequency of anemia in patients with infarcts in multiple vascular territories was significantly higher than in patients with infarcts in single vascular territory.

## 5. Limitations

The most important limitations of the study include the use of data from a single center and data from records, as well as the relatively small sample size. Current guidelines recommend rhythm Holter monitorization for more than 24 hours in patients with cryptogenic stroke. However, this proposal could not be complied with due to the retrospective nature of our study. Our study will pioneer further multicenter, prospective studies with longer rhythm Holter monitorization, investigating the relation between premature atrial contractions, short atrial run attacks, and cryptogenic stoke.

## 6. Conclusion

Accurate identification of the etiological cause in ischemic stroke patients is important because it modifies antiaggregant or anticoagulant treatment options. Identifying new etiologic factors for this disease, which is one of the leading causes of mortality and disability in the community, will be important in guiding both primer and secondary prevention methods. Therefore, if frequent premature atrial contractions and/or short atrial run attacks are monitored in the patient’s rhythm Holter monitoring before the diagnosis of cryptogenic stroke, more effort should be spent to detect a possible cardioembolic etiology, and a rhythm Holter follow-up should be performed as long as possible.

## Figures and Tables

**Figure 1 medicina-55-00038-f001:**
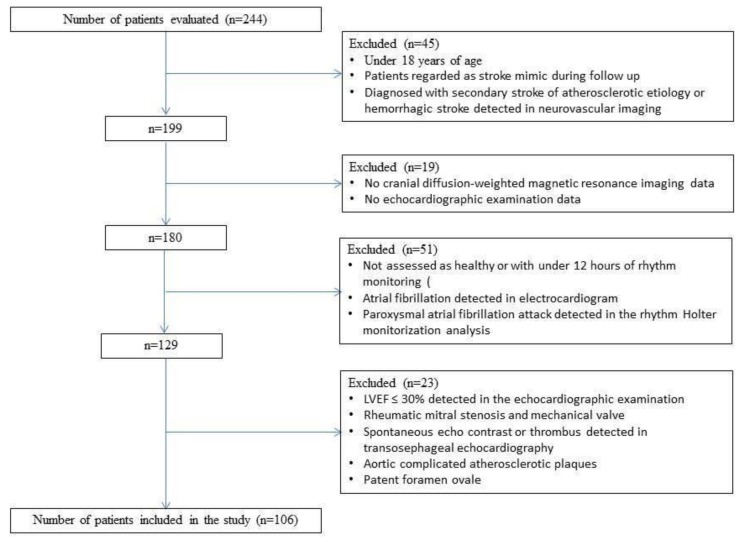
Inclusion and exclusion criteria.

**Figure 2 medicina-55-00038-f002:**
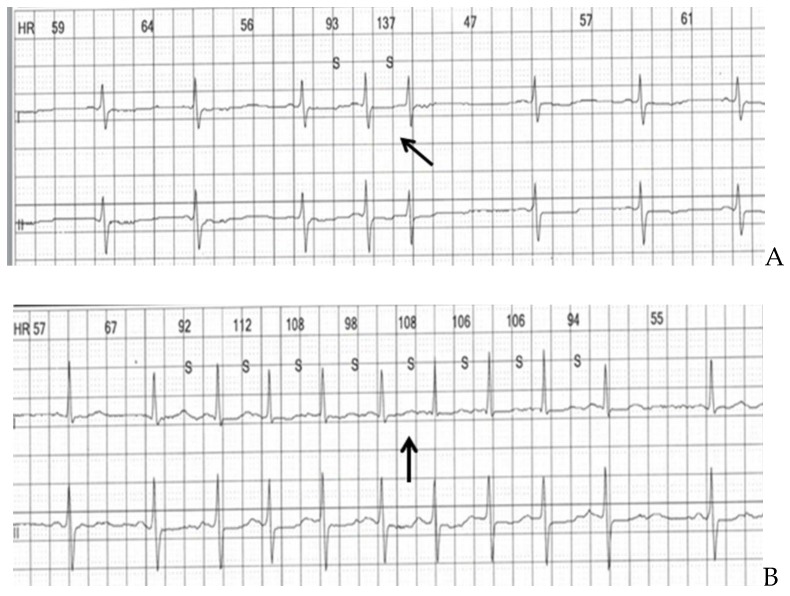
(**A**) Representation of premature atrial contraction from the rhythm Holter analysis of a patient with infarcts in multiple vascular territories. (**B**) Representation of 8-beat supraventricular run from the rhythm Holter analysis of a patient with infarcts in multiple vascular territories.

**Figure 3 medicina-55-00038-f003:**
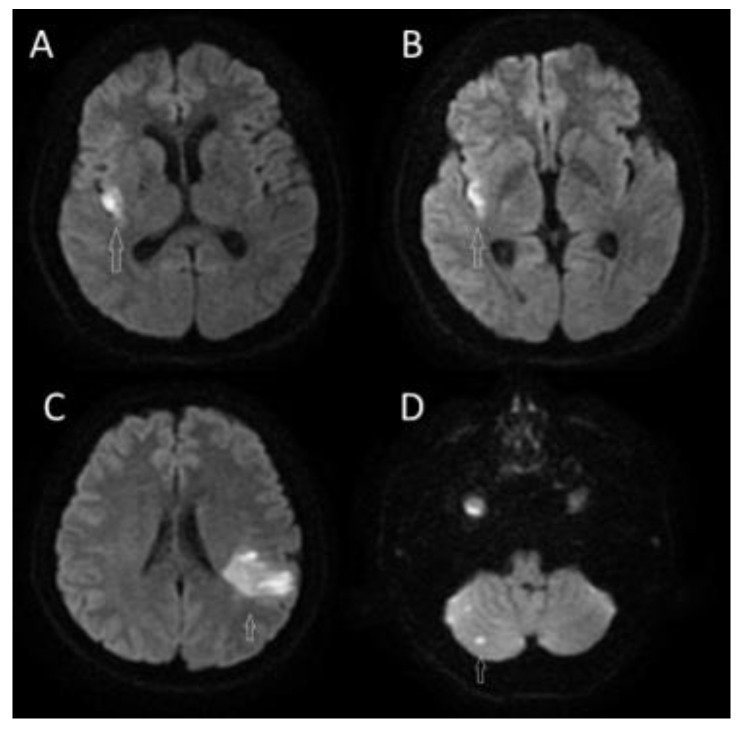
(**A**) and (**B**) single vascular territory infarction, (**C**) and (**D**) multiple vascular territory infarction.

**Table 1 medicina-55-00038-t001:** Demographic data between single and multiple vascular territory infarcts.

Variables	Single Vascular Territory (n = 73)	Multiple Vascular Territory (n = 33)	*p* value
**Age**	59.07 ± 14.21	65.12 ± 12.79	**0.039***
**Gender**			
Female	21 (61.8)	13 (38.2)	0.278
Male	52 (72.2)	20 (27.8)	
**Alcohol**			
No	62 (68.1)	29 (31.9)	0.773
Yes	11 (73.3)	4 (26.7)	
**Smoking**			
No	46 (64.8)	25 (35.2)	0.196
Yes	27 (77.1)	8 (22.9)	
**HT**			
No	40 (78.4)	11 (21.6)	**0.041***
Yes	33 (60.0)	22 (40.0)	
**DM**			
No	54 (74.0)	19 (26.0)	0.091
Yes	19 (57.6)	14 (42.4)	
**HL**			
No	50 (74.6)	17 (25.4)	0.093
Yes	23 (59.0)	16 (41.0)	
**CAD**			
No	55 (71.4)	22 (28.6)	0.354
Yes	18 (62.1)	11 (37.9)	

* Statistical significance (*p* < 0.05); descriptives: mean±SD, count (percent). Abbreviations: CAD: Coronary Artery Disease, DM: Diabetes Mellitus, HL: Hyperlipidemia, HT: Hypertension.

**Table 2 medicina-55-00038-t002:** Distribution of laboratory data variables of patients.

Variables	Single Vascular Territory (n = 73)	Multiple Vascular Territory (n = 33)	*p* value
Rhythm Holter Duration	23.15 (22.00–24.00)	23.18 (21.16–24.00)	0.795
Maximum Heart Rate	113.63 ± 18.91	112.18 ± 19.21	0.3633
Mean Heart Rate	71.9589 ± 8.8858	75.36 ± 12.52	0.113
Minimum Heart Rate	50.51 ± 8.23	52.97 ± 12.75	0.236
Total PAC Count	167.00 (37.00–447.00)	1196.00 (1061.00–1813.00)	<0.001*
Atrial Run	0.00 (0.00–0.00)	13.50 (11.00–34.25)	<0.001*
Total Run	11.42 (11.31–11.54)	11.38 (11.31–11.53)	0.286
PAC/Total heart rate	0.0018 (0.0004–0.0051)	0.0174 (0.0098–0.0275)	<0.001*
EF	60.00 (55.00–63.00)	58.00 (55.00–60.00)	0.337
LA	36.99 ± 6.07	37.09 ± 5.26	0.932
LVDD			
No	29 (72.5)	11 (27.5)	0.530
Yes	44 (66.7)	22 (33.3)	
CHA2DS2-VASc	3.00 (2.00–5.00)	5.00 (3.00–6.00)	0.059
logPLT	12.36±0.45	12.39 ± 0.30	0.873
logWBC	8.96 ± 0.26	8.98 ± 0.35	0.790
CRP	0.60 (0.30–1.06)	1.00 (0.40–3.20)	0.037*
TSH	0.80 (0.40–1.20)	1.00 (0.40–1.90)	0.479
HG	14.00 (12.70–15.00)	13.00 (11.30–13.70)	<0.001*
MPV	10.00 (9.20–10.90)	9.60 (9.20–10.30)	0.305

* Statistical significance (*p* < 0.05); descriptives: mean ± sd, median (25th––75th percentile), count (percent). Abbreviations: CRP: C-reactive protein, HG: Hemoglobin, EF: Ejection fraction, LA: Left atrium, LVDD: Left ventricular diastolic dysfunction, MPV: Mean platelet volume, PAC: Premature atrial contraction PLT: Platelet WBC: White blood cell, TSH: Thyroid-stimulating hormone. CHA2DS2-VASc nomenclature represents heart failure (C), hypertension (H), age ≥ 75 years (A2), diabetes mellitus (D), stroke (S2), vascular disease (V), age 65 to 74 years (A) and female gender (as a sex category [Sc]).

**Table 3 medicina-55-00038-t003:** Univariate logistic regression analyses results for single and multiple vascular territories.

Variable	Coefficient (Standard Error)	Odds Ratio (95% CI)	*p* value
Age	0.034 (0.017)	1.034 (1.001–1.068)	**0.042***
Gender (Female)	0.476 (0.440)	1.610 (0.679–3.814)	0.280
Alcohol (Yes)	−0.252 (0.626)	0.777 (0.228–2.650)	0.687
Smoking (Yes)	−0.607 (0.473)	0.545 (0.216–1.378)	0.200
HT (Yes)	0.886 (0.438)	2.424 (1.028–5.718)	**0.043***
DM (Yes)	0.739 (0.442)	2.094 (0.881–4.978)	0.094*
HL (Yes)	0.716 (0.430)	2.046 (0.881–4.752)	0.096*
CAD (Yes)	0.424 (0.458)	1.528 (0.622–3.752)	0.355
Rhythm Holter Duration	0.034 (0.075)	1.034 (0.893–1.198)	0.652
Maximum Heart Rate	−0.004 (0.011)	0.996 (0.974–1.018)	0.714
Mean Heart Rate	0.033 (0.021)	1.033 (0.992–1.076)	0.117*
Minimum Heart Rate	0.026 (0.022)	1.026 (0.983–1.070)	0.238
Total PAC Count	0.003 (0.001)	1.003 (1.002–1.004)	**<0.001***
Atrial Run	0.119 (0.030)	1.126 (1.062–1.195)	**<0.001***
Total Run	−0.000012 (0.000010)	1.000 (0.999–1.000)	0.250
EF	-0.016 (0.030)	0.984 (0.927–1.044)	0.586
LA	0.003 (0.036)	1.003 (0.934–1.077)	0.931
LVDD (Yes)	0.276 (0.440)	1.318 (0.557–3.122)	0.530
CHA2DS2-VASc	0.220 (0.116)	1.246 (0.993–1.565)	**0.058***
logPLT	0.232 (0.553)	1.261 (0.426–3.729)	0.675
logWBC	0.194 (0.720)	1.214 (0.296–4.978)	0.787
CRP	0.162 (0.076)	1.175 (1.014–1.363)	**0.032***
TSH	0.174 (0.145)	1.190 (0.896–1.581)	0.230
HG	−0.373 (0.127)	0.689 (0.537–0.884)	**0.003***
MPV	−0.124 (0.174)	0.884 (0.629–1.242)	0.477

* *p* < 0.20. Abbreviations: CI: Confidence interval, CAD: Coronary artery disease, CRP: C-reactive protein, DM: Diabetes mellitus EF: Ejection fraction, HG: Hemoglobin, HL: Hyperlipidemia, HT: Hypertension, LA: Left atrium, LVDD: Left ventricular diastolic dysfunction, MPV: Mean platelet volume, PAC: Premature atrial contraction PLT: Platelet, WBC: White blood cell, TSH: Thyroid-stimulating hormone.

**Table 4 medicina-55-00038-t004:** Multivariate logistic regression analysis result for single and multiple vascular territories.

Variable	Coefficient (Standard Error)	Odds Ratio (95% CI)	*p* value
Age	0.034 (0.041)	1.035 (0.956–1.121)	0.396
HT (Yes)	0.028 (1.145)	1.028 (0.109–9.706)	0.980
DM (Yes)	1.529 (0.994)	4.612 (0.657–32.382)	0.124
HL (Yes)	0.903 (0.981)	2.467 (0.360–16.883)	0.358
Mean Heart Rate	0.020 (0.040)	1.020 (0.942–1.104)	0.626
Total PAC Count	0.002 (0.001)	1.002 (1.001–1.004)	0.001*
Atrial Run	0.083 (0.031)	1.086 (1.021–1.155)	0.008*
CHA2DS2-VASc	−0.872 (0.499)	0.418 (0.157–1.112)	0.080
CRP	0.094 (0.097)	1.099 (0.909–1.328)	0.332
HG	−0.286 (0.214)	0.751 (0.494–1.142)	0.180

* *p* < 0.05. Abbreviations: CI: Confidence interval, CRP: C-reactive protein, DM: Diabetes mellitus, HL: Hyperlipidemia, HT: Hypertension, PAC: Premature atrial contraction.

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
