# Peer review of "The Relationship between Diffusion-Weighted Magnetic Resonance Imaging Lesions and 24-Hour Rhythm Holter Findings in Patients with Cryptogenic Stroke"

_1010-660X, 2019, doi:10.3390/medicina55020038_

Round 1

Reviewer 1 Report

This manuscript by Gürdogan et al from an unknown centre reports a retrospective study, which associates premature atrial contractions and atrial runs with the presence of uniterritorial stroke of unknown origin (compared with multiterritorial stroke). To me the study is very interesting, as it proves the notion that embolic stroke of unknown source (ESUS) may develop from undetected paroxysmal atrial fibrillation. However, the study is too small and includes limited resources (Holter recordings for only 23 hours) so that it will not change clinical practice. Additionally, I have some comments/suggestions:

·        Was any power calculation performed?

·        New lines between individual sections of the abstract, as well as complete affiliations would be favourable.

·        The listing of differences between multi-territory and single-territory infarction is useless in the abstract section if the direction of differences is not highlighted. For example: Is LOW or HIGH age correlated with multiple territories? The same is true for CRP, haemoglobin, premature atrial contractions, short atrial run count and the presence of hypertension.

·        Please present excluded patients in a study flowchart and present the numbers in the Results section. Define clear inclusion and exclusion criteria in the Methods section (they are currently spread in chapters 2.1 and 2.3).

·        Please remove the second sentence (“patients were divided…”) and the forth sentence (“The difference in demographic…”) in the results section as these sentences are redundant and belong to the Methods.

·        Please put the percent sign (%) at the end of each number.

·        Please replace “Table 3” with “Table 4” in line 144.

·        Line 205: “Gladstone and colleagues…”: this sentence makes no sense.

Author Response

Upon your request for revision, I am resubmitting the Manuscript ID medicina-424665" The Relationship Between Diffusion-Weighted Magnetic Resonance Imaging Lesions and 24-Hour Rhythm Holter Findings in Patients With Cryptogenic Stroke  ".Responses to the Reviewers are available below, and changes have been indicated in red in the manuscript.

With best regards,

Muhammet Gürdoğan, MD

Reviewers' Comments to Author:

Reviewer 1:

This manuscript by Gürdogan et al from an unknown centre reports a retrospective study, which associates premature atrial contractions and atrial runs with the presence of uniterritorial stroke of unknown origin (compared with multiterritorial stroke). To me the study is very interesting, as it proves the notion that embolic stroke of unknown source (ESUS) may develop from undetected paroxysmal atrial fibrillation. However, the study is too small and includes limited resources (Holter recordings for only 23 hours) so that it will not change clinical practice. Additionally, I have some comments/suggestions:

•        Was any power calculation performed?

We performed the power analysis and we added it to the Statistical Analysis section.

•        New lines between individual sections of the abstract, as well as complete affiliations would be favourable.

The Abstract section is re-edited.

•        The listing of differences between multi-territory and single-territory infarction is useless in the abstract section if the direction of differences is not highlighted. For example: Is LOW or HIGH age correlated with multiple territories? The same is true for CRP, haemoglobin, premature atrial contractions, short atrial run count and the presence of hypertension.

We removed this sentence from the Abstract section upon your wish:  “Significant differences were found between two groups in age (p=0.042), CRP (p=0.032), hemoglobin (p=0.003), total premature atrial contraction count (p<0.001), short atrial run count (p<0.001), and presence of hypertension (p=0.043).”

•        Please present excluded patients in a study flowchart and present the numbers in the Results section. Define clear inclusion and exclusion criteria in the Methods section (they are currently spread in chapters 2.1 and 2.3).

We defined inclusion and exclusion criteria in the Methods section, and we added to flowchart Results section.

•        Please remove the second sentence (“patients were divided…”) and the forth sentence (“The difference in demographic…”) in the results section as these sentences are redundant and belong to the Methods.

We removed these sentences.

•        Please put the percent sign (%) at the end of each number.

We changed the place of the percent signs (%) in Table 1 and 2.

•        Please replace “Table 3” with “Table 4” in line 144.

We changed the expression “Table 3” as “Table 4” in 144.

•        Line 205: “Gladstone and colleagues…”: this sentence makes no sense.

We changed the sentence as:  “Gladstone et al., reported that a significant number of more atrial fibrillation episodes could be detected with noninvasive thirty days monitoring in patients with cryptogenic stroke or transient ischemic attacks, and thus started oral anticoagulant therapy twice.”

Reviewer 2 Report

In this article, Gurdogan et al. aimed to investigate the correlation between infarct distribution of patients with cryptogenic stroke and the rhythm Holter monitoring data of the patients.

Overall, this is a focused and organized clinical data analysis study. By using multiple statistical analysis, the authors revealed the data collected from premature atrial contraction and short atrial run could separate single and multiple circulation infarcts. While the paper provided a potential method to identify the etiological cause in ischemic stroke patients by using rhythm Holter monitor, this manuscript needs to be revised based on the following concerns:

Major:

1.       The part of introduction was lack of related background. Additionally, the authors should make a clear conclusion based on the results rather than prospect.

2.       The authors preferred long sentences. But some of them were too long to understand. Please maintain an appropriate length of your sentence so that you can convey your message more effectively to the readers.

Minor:

1.       The description in page 1 line 12-16 was hard to follow. Please revise the long sentence and make it understandable.

2.       In page 4 line 123-129 the author should cite Table 2 while describing the data.

3.       The author used the term infarcts in single/multiple vascular territory in the result section while used single/multiple circulation infarcts in the method and table. It’s better to use consistent term to describe the same objective.

4.       In table 2, the number of CHADVASC, CRP, TSH, HG and MFV were listed in the format of number (the range of numbers), which was obvious neither mean±SD nor N(%) claimed by the author. The author should indicate what the number represents.

Please interpret what %95 GA stands for in table 3 and 4.

Author Response

Upon your request for revision, I am resubmitting the Manuscript ID medicina-424665" The Relationship Between Diffusion-Weighted Magnetic Resonance Imaging Lesions and 24-Hour Rhythm Holter Findings in Patients With Cryptogenic Stroke  ".Responses to the Reviewers are available below, and changes have been indicated in red in the manuscript.

With best regards,

Muhammet Gürdoğan, MD

Reviewers' Comments to Author:

Reviewer 2:

In this article, Gurdogan et al. aimed to investigate the correlation between infarct distribution of patients with cryptogenic stroke and the rhythm Holter monitoring data of the patients.

Overall, this is a focused and organized clinical data analysis study. By using multiple statistical analysis, the authors revealed the data collected from premature atrial contraction and short atrial run could separate single and multiple circulation infarcts. While the paper provided a potential method to identify the etiological cause in ischemic stroke patients by using rhythm Holter monitor, this manuscript needs to be revised based on the following concerns:

Major:

1.       The part of introduction was lack of related background. Additionally, the authors should make a clear conclusion based on the results rather than prospect.

We revised the Introduction section by adding extra expressions.

2.       The authors preferred long sentences. But some of them were too long to understand. Please maintain an appropriate length of your sentence so that you can convey your message more effectively to the readers.

We re-edited long sentences by dividing them in short sentences in the text.

 Minor:

1.       The description in page 1 line 12-16 was hard to follow. Please revise the long sentence and make it understandable.

We changed the sentence as: This study was conducted to investigate the relationship            between short atrial runs and frequent premature atrial contractions  detected in Holter monitors and  infarct distributions in cranial magnetic resonance imaging of patients diagnosed with cryptogenic stroke.”

2.       In page 4 line 123-129 the author should cite Table 2 while describing the data.

We added this section to Table information.

3.       The author used the term infarcts in single/multiple vascular territory in the result section while used single/multiple circulation infarcts in the method and table. It’s better to use consistent term to describe the same objective.

Vascular territory term was preferred in the whole text.

4.       In table 2, the number of CHADVASC, CRP, TSH, HG and MFV were listed in the format of number (the range of numbers), which was obvious neither mean±SD nor N(%) claimed by the author. The author should indicate what the number represents.

We added the Specified definitions below Table 2 such as descriptives: mean±sd, median (25th percentile – 75th percentile), count (percent). 

      Please interpret what %95 GA stands for in table 3 and 4.

         We changed “%95 GA” as “95% CI” (Confidence Interval).
